# The Impact of Missing Data and Imputation Methods on the Analysis of 24-Hour Activity Patterns

**Lara Weed [1], Renske Lok [2], Dwijen Chawra [2] and Jamie Zeitzer [2,3,*]**

[1] Department of Bioengineering, Stanford University, Stanford, CA 94305, USA
[2] Department of Psychiatry and Behavioral Sciences, Stanford University, Stanford, CA 94305, USA
[3] Mental Illness Research Education and Clinical Center, VA Palo Alto Health Care System, Palo Alto, CA 94304, USA
[*] Correspondence: jzeitzer@stanford.edu

**Abstract:** The purpose of this study is to characterize the impact of the timing and duration of missing actigraphy data on interdaily stability (IS) and intradaily variability (IV) calculation. The performance of three missing data imputation methods (linear interpolation, mean time of day (ToD), and median ToD imputation) for estimating IV and IS was also tested. Week-long actigraphy records with no non-wear or missing timeseries data were masked with zeros or 'Not a Number' (NaN) across a range of timings and durations for single and multiple missing data bouts. IV and IS were calculated for true, masked, and imputed (i.e., linear interpolation, mean ToD and, median ToD imputation) timeseries data and used to generate Bland–Alman plots for each condition. Heatmaps were used to analyze the impact of timings and durations of and between bouts. Simulated missing data produced deviations in IV and IS for longer durations, midday crossings, and during similar timing on consecutive days. Median ToD imputation produced the least deviation among the imputation methods. Median ToD imputation is recommended to recapitulate IV and IS under missing data conditions for less than 24 h.

**Keywords:** actigraphy; circadian rhythms; interdaily stability; intradaily variability; imputation

## 1. Introduction

In recent years, the use of wearable sensors for remote and longitudinal monitoring has increased in prevalence across multiple disciplines. While wearables have decreased in size and increased in battery life, current form factors still suffer from spurious or missing data due to removal by users [1,2]. Spurious data due to non-wear (typically repeated zero-values) can lead to unreliable results in some algorithms (e.g., mistaking non-wear for sleep) and consequently, various methods for detecting non-wear have been developed [3,4]. Similarly, bouts of missing data may limit accurate assessment. In general, the threshold for tolerable amounts of spurious or missing timeseries data and algorithmic methods for minimizing its impact through imputation has not been explicitly explored for many applications.

In the field of sleep and circadian rhythms, accelerometry recorded from the wrist (actigraphy) is commonly used to study ambulatory sleep-wake and activity patterns [5,6]. While some algorithms used to examine daily activity patterns may be more robust to missing data, such as cosinor analysis [7], they often rely on underlying pattern matching assumptions, which may not extend to populations with sleep-wake disturbances or deviating activity patterns [8–10]. Nonparametric algorithms can quantitate activity patterns without *a priori* assumptions about the shape of the activity patterns.

A set of commonly used nonparametric metrics is that of intradaily variability (IV) and interdaily stability (IS). IV characterizes the average degree of hour-to-hour activity variability within a day, and IS characterizes the regularity of hourly activity between days. IV and IS have been examined in many hundreds of manuscripts and differences

in these measures are associated with the severity and time course of a variety of disease processes, including bipolar disorder, schizophrenia, and depression, among others [10–17]. However, using at least 5 days of continuous data without non-wear or missing periods is recommended to make reliable estimations [18]. Moreover, based on the mathematical structure of the calculations, these metrics are more sensitive to missing data than cosinor analyses. The amount and timing of missing data and its relative impact on non-parametric measures such as IV and IS is not well understood [19].

Several methods of timeseries imputation have been used to fill non-wear and missing actigraphy data ranging from methods relying exclusively on data surrounding the gap such as simple linear interpolation [20,21], methods relying on data from other days during the same time of day such as time-of-day-based mean and median imputation [19,21], and more sophisticated approaches leveraging larger datasets such as deep learning methods [21]. However, the impact of the method used to impute the data, especially in the context of long bouts of consecutive missing data, on the calculation of IV and IS is unknown.

The purpose of this study is two-fold: (1) to determine the impact of the two missing data phenotypes (i.e., spurious zero-values and missing "Not a Number" (NaN) values), both in duration and clock time, on the calculation of IV and IS and (2) to determine the utility of different imputation methods in replacing missing data when calculating IV and IS. To accomplish this, we examined data obtained from the UK Biobank, a large, community-based sample of adults in the United Kingdom.

## 2. Results

### 2.1. Participant Characteristics

The subset of n = 84 individuals randomly selected from the UK Biobank dataset pool containing no identified non-wear were about half female (n = 47) and predominantly white (n = 83). The age range during accelerometry data collection was diverse, with the youngest and oldest included individuals being 47 and 77 years old, respectively (median [IQR]: 64 [56–67] years). Townsend Deprivation Index, a measure of material deprivation within a population with higher scores representing higher material deprivation, ranged from −6.18 to 4.69 (median [IQR]: −2.2 [−3.75–0.25]). IV ranged from 0.46 to 1.62 (median [IRQ]: 0.91 [0.78–1.06]. IS ranged from 0.15 to 0.80 (median [IRQ]: 0.54 [0.44–0.63]).

### 2.2. Single Gap Imputation—IS

Simulation of missing timeseries data via masking across various durations and starting times of day indicates that missing data primarily impacts the mean difference in IS (Figure 1). When masked with zeros compared to complete data, IS mainly becomes artificially lower (Figures 1E and S2), has a moderate increase in standard deviation (Figure S3). An increase in magnitude of the slope, especially during the overnight, indicating that IS estimation was systematically worse with smaller values of IS (Figure S4). When masked with NaNs compared to complete data, IS mainly becomes artificially higher (Figures 1D and S2), has a low increase in standard deviation (Figure S3), and has little effect on slope (Figure S4). Longer durations of masked data as well as mid-morning starts had the largest impact on IS with lower values reported for masking with zero and higher values reported for masking with NaNs (Figure 1D–E). Linear interpolation was generally poor at recapitulating true IS (Figure 1A). While it reduced the error for missing data durations less than approximately 7 h (Figure 1A), it did not do so for longer durations at many times of day. Linear interpolation also worsened the standard deviation (Figure S3) and slope (Figure S4) calculations for missing data durations longer than 7 h. Mean time-of-day (ToD) imputation allowed for recapitulation of most IS values except for long duration data gaps that started during the night (Figure 1B). Mean ToD imputation kept both standard deviation (Figure S3) and slope (Figure S4) relatively low. Median ToD imputation had the best results in that the imputed data led to IS that were within 0.05 units of actual IS values

(Figure 1C) and, as with mean imputation, kept both the standard deviation (Figure S3) and the slope (Figure S4) relatively low.

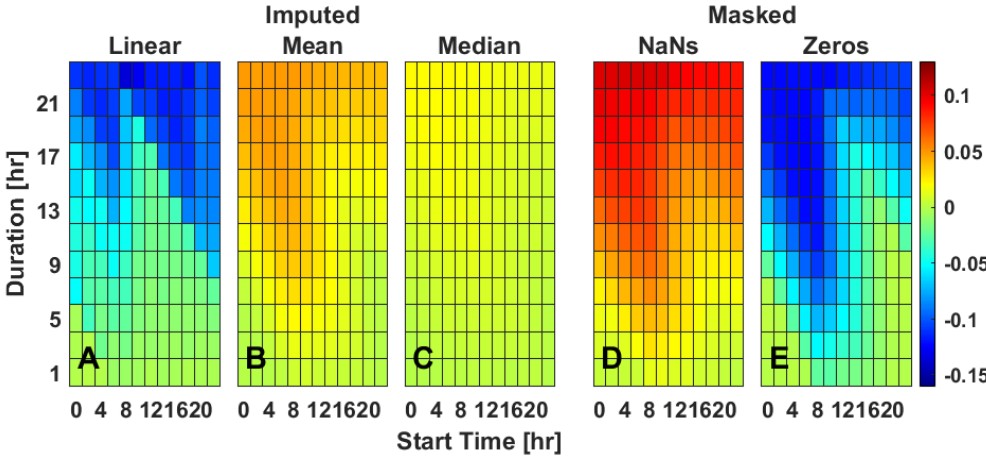

**Figure 1.** Rhythm regularity (IS) for a single missing data gap starting on a representative day (Tuesday). Data are the mean difference between the masked and true IS values (**D**,**E**) or imputed and true IS values (**A**–**C**), as extracted from Bland–Altman plots. Three different imputation methods [linear interpolation (**A**), mean Time of day (ToD) (**B**), median ToD (**C**)] and two masking methods [NaNs (**D**), zeros (**E**)] are presented for varied durations (*y*-axis) and timing (*x*-axis) of masked data gaps. Values are color-coded as indicated with best performance being closer to 0 (green). For heat maps of each individual day of rhythm regularity, see Supplemental Data Figure S2.

### 2.3. Single Gap Imputation—IV

Simulation of missing data via masking across various durations and starting times of day indicates that missing data primarily impacts the mean difference in IV for data masked with zeros but not for data masked with NaNs. When masked with zeros compared to the complete dataset, IV becomes artificially both lower and higher than would have been calculated, especially with data gaps longer than 13 h (Figures 2E and S5). When masked with NaNs compared to the complete dataset, IV showed little deviation in mean (Figures 2D and S5). Masking had a moderate impact on increasing the standard deviation in the difference between IV calculated from true and masked datasets (Figure S6). Masking did not, however, have a large impact on the slope (Figure S7), indicating that the relationship between IV calculated from complete and missing datasets did not systematically vary based on the magnitude of IV. Linear interpolation (Figure 2A) was generally poor at recapitulating true IV. For most durations and times of day, linear interpolation of missing data made IV less accurate than if the data were masked with zeros, especially at durations longer than approximately 13 h (Figure 2A). Linear interpolation also worsened the standard deviation (Figure S6) and slope (Figure S7) calculations for many combinations of missing data durations and start times. Both mean ToD (Figure 2B) and median ToD (Figure 2C) imputation similarly corrected errors in IV due to missing data, though neither decreased the standard deviation error (Figure S6).

### 2.4. Multiple Gap Imputation—IS

While a single short period of missing data has relatively little impact on the calculation of IS (Figure 1D,E), multiple bouts of short (115 min, 140 min, Figure 1) missing data segments impacted IS when masked with zeros but not when masked with NaNs (Figures 3D,E and S8). When masking with zeros compared to complete IS data, the greatest deviations from the mean occurred with a banding pattern, indicating that missing data during the same time on consecutive days affects IS scores most (Figure 3E). Standard deviation has a moderate increase in both masking conditions. Linear interpolation and mean ToD and median ToD imputation methods each performed well and similarly across

mean and slope measures (Figures 3, S8 and S10). Standard deviation was elevated for linear interpolation compared to other imputation methods and masking (Figure S9).

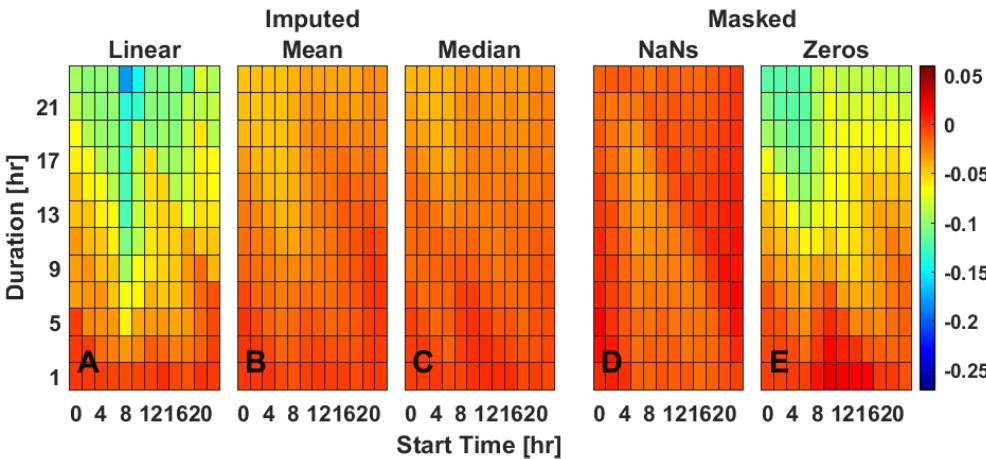

**Figure 2.** Rhythm fragmentation (IV) for a single missing data gap starting on a representative day (Tuesday). Data are the mean difference between the masked and true IV values (**D**,**E**) or imputed and true IV values (**A**–**C**) as extracted from Bland–Altman plots. Three different imputation methods [linear interpolation (**A**), mean ToD (**B**), median ToD (**C**)] and two masking methods [NaNs (**D**), zeros (**E**)] are presented for varied durations (*y*-axis) and timing (*x*-axis) of masked data gaps. Values are color-coded as indicated with best performance being closer to 0. For heat maps of each individual day of rhythm fragmentation, see Supplemental Data Figure S5.

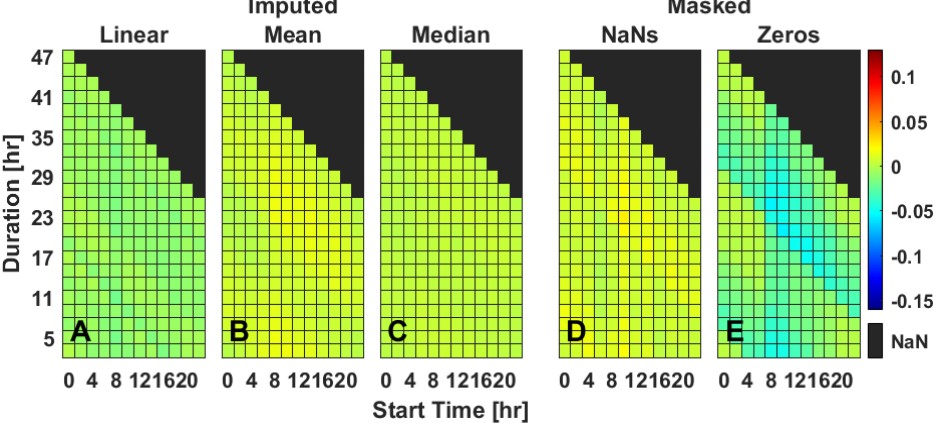

**Figure 3.** Rhythm regularity (IS) for two gaps (gap 1: 115 min, gap 2: 140 min) of missing data starting on a representative day (Tuesday). Data are the mean difference between the masked and true IS values (**D**,**E**) or imputed and true IS values (**A**–**C**), as extracted from Bland–Altman plots. Three different imputation methods [linear interpolation (**A**), mean ToD (**B**), median ToD (**C**)] and two masking methods [NaNs (**D**), zeros (**E**)] are presented for varied durations between bouts (*y*-axis) and timings (*x*-axis) of masked data gaps. Values are color-coded as indicated with best performance being closer to 0; NaN values indicate where values could not be calculated due to dataset constraints. For heat maps of each individual day of rhythm regularity, see Supplemental Data Figure S8.

## 2.5. Multiple Gap Imputation—IV

Masked data predominantly affected IV score with at least one gap of missing data at midday, in a banding pattern (Figure 4D,E). Linear, mean, and median imputation methods each performed well and similarly across mean, standard deviation, and slope (Figures 4 and S11–S13).

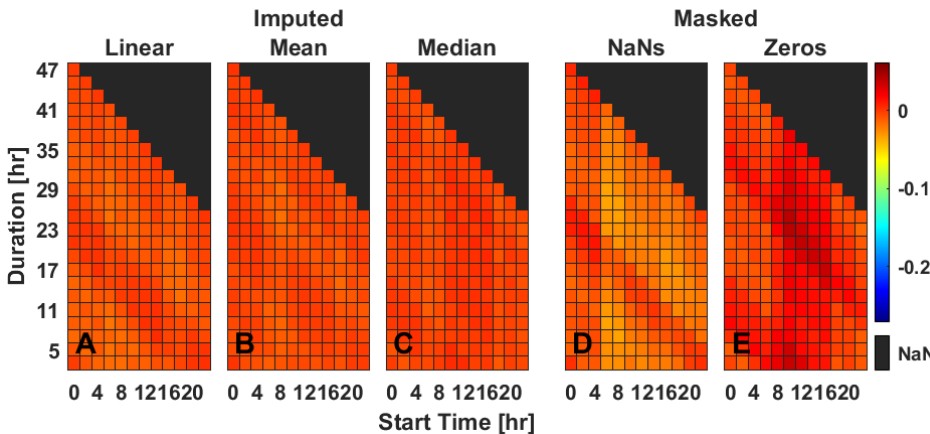

**Figure 4.** Rhythm fragmentation (IV) for two gaps (gap 1:115 min, gap 2:140 min) of missing data starting on a representative day (Tuesday). Data are the mean difference between the masked and true IV values (**D**,**E**) or imputed and true IV values (**A**–**C**), as extracted from Bland–Altman plots. Three different imputation methods [linear interpolation (**A**), mean ToD (**B**), median ToD (**C**)] and two masking methods [NaNs (**D**), zeros (**E**)] are presented for varied durations between bouts (*y*-axis) and timing (*x*-axis) of masked data gaps. Values are color-coded as indicated with best performance being closer to 0; NaN values indicate where values could not be calculated due to dataset constraints. For heat maps of each individual day of rhythm fragmentation, see Supplemental Data Figure S11.

## 3. Discussion

Our results suggest that IS and IV are most sensitive to missing data with start times midday and morning, respectively, and both are sensitive to longer missing data durations. The magnitude of the impact of missing data is not insubstantial, being similar to the differences that have been observed between controls and a variety of populations, including those with either unipolar or bipolar depression [11,12]. Thus, failure to accurately account for missing data can lead to inappropriate conclusions, especially if there is an expectation of differential data loss between two populations (i.e., one more likely to not wear the activity recorder). However, imputing periods of missing data with the median acceleration measured at other days at those times of day can adequately replace data loss up to 24 h and recapitulate expected IS and IV calculated from a full week of data.

While both IS and IV are impacted by non-wear data phenotypes (i.e., masking with zeros), sensor failure phenotypes (i.e., masking with NaNs) has relatively little impact on IV for data loss up to 24 h. The reason that IV is more robust to missing data as compared to IS may be due to the formulation of these calculations and their normalization. For IV, the raw data for a week are collapsed into N-1 terms representing the number of hours in a week minus one (i.e., 167), whereas for IS the data are collapsed into p terms, representing the number of hours in a day (i.e., 24). Due to this, one hour of missing data will impact one out of 167 terms (0.6%) in the calculation of IV and one out of 24 terms (4.2%) in the calculation of IS. This also indicates that these metrics are more sensitive to spurious, non-wear data than sensor failure phenotypes indicating the importance of non-wear detection.

Timing of missing data also had an impact with midday gaps affecting results at shorter durations. This may be due to the typical patterns of human activity with highest activity levels and day-to-day variability generally occurring midday. Missing data during this timeframe would therefore have a greater impact compared to other timeframes on both the calculation of IV and IS. This is also consistent with the finding that for multiple gaps, similar times on consecutive days and midday crossovers had the greatest impact on IS, whereas IV was particularly sensitive to midday gaps.

The imputation methods selected here are statistical and are not meant to recreate the missing timeseries data but to improve the accuracy of IV and IS calculation. The simplest methods of imputation were initially selected to explore this application. Other imputation methods could have been chosen and may very well improve upon the results

observed here, however, the median imputation method, and to a lesser extent the mean imputation method, is sufficient to replace most missing data <24-h duration. We speculate that median imputation was less susceptible to outliers and captured more signal variability than median imputation. We did not impute data that were shorter than one hour, though our results indicating that intentionally masking for two hours has relatively little impact on IV and IS imply that a 1 h data loss would have minimal impact on IV and IS and does not need to be detected or imputed for such calculations.

We tested these algorithms in a randomly selected population of community dwelling individuals using a specific actigraph (Axivity). While it is unlikely that the choice of monitor would change the implications of these findings, given the continued small impact of data loss on slope even after imputation, it is possible that populations in which there is an expectation for less consolidated or more irregular activity could benefit more from a different imputation method.

Overall, IV and IS measured from wrist actigraphy is sensitive to both known and unknown missing data. Median ToD imputation is capable of recapitulating IV and IS values under missing data conditions for up to 24 h from a week-long recording. Future studies should explore the stability of IV and IS with variable recording durations.

## 4. Materials and Methods

### 4.1. Dataset

Data were obtained from the UK Biobank database (project ID 63099), a large-scale biomedical research resource with >500,000 participants recruited from the general population of England, Wales, and Scotland, aged between 40 and 69 years in 2006–2010 [22]. Between 2013 and 2015, a subset (n = 103,685) of individuals participated in wrist actigraphy data collection. Participants were asked to wear a wrist actigraph (AX3, Axivity, Newcastle upon Tyne, UK) for one week. The device, similar to a standard fitness tracker, is equipped with a tri-axial accelerometer recording at 100 Hz with a dynamic range of $\pm 8$ g. We excluded participants who withdrew, had unreliable data or calibration, wear durations as defined by UK Biobank of shorter than 5 days, or recordings during the Daylight Savings time switch or the week following. We also excluded participants with non-wear periods spanning multiple days, leaving a final sample of 83,937 participants (Figure 5A). These data were down-sampled to a single vector magnitude value with noise and gravity removed for every 30 s interval (biobank accelerometer analysis, Python 3.6.1) [23]. A subset (0.01%, n = 84) of individuals with at least 7 days of data and no bouts of non-wear (588 total days) were randomly selected for further analysis of the impact of missing data on the calculation of IV and IS (Figure 5B).

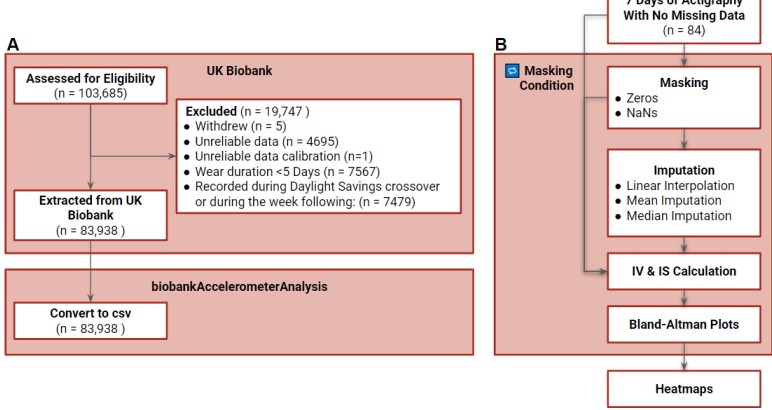

**Figure 5.** Consort Diagram. In total, 103,685 files were assessed for eligibility, of which 19,747 were excluded, resulting in 83,938 accelerometer files (**A**). A random subset (N = 84 files, 0.01% of extracted sample) of individuals with at least 7 days of data without missing data were subjected to masking, imputation, IV and IS calculation (**B**).

### 4.2. Calculation of IV and IS

IV and IS are common metrics calculated in the assessment of activity patterns spanning multiple days. Both metrics leverage hourly average activity and hour-to-hour changes in activity levels to characterize patterns of activity. IV quantifies the degree of consolidation of activity by calculating the normalized ratio of the sum of the squared hour-to-hour changes in activity to the sum of the squared difference in hourly activity from the overall average across the data, and is calculated as:

$$\text{IV} = \frac{n \sum_{i=2}^{n} (X_i - X_{i-1})^2}{(n-1) \sum_{i=1}^{n} (X_i - \overline{X})^2} \tag{1}$$

where $n$ is the total number of hours in the data collection (168 h for 7 days of data), $X_i$ is the hourly average at hour $i$, and $\overline{X}$ is the average across all hours. IV values range from 0 to 2, with lower values representing greater consolidation.

IS quantifies the degree of stability of the hourly activity pattern between days by calculating the normalized ratio of the sum of the squared difference in average activity from each hour of the day from overall average activity to the sum of the squared difference in hourly activity from the overall average across the data, and is calculated as:

$$\text{IS} = \frac{n \sum_{h=1}^{p} (\overline{X_h} - \overline{X})^2}{p \sum_{i=1}^{n} (X_i - \overline{X})^2} \tag{2}$$

where $p$ is the hour of the day ranging from 1 to 24 and $\overline{X_h}$ is the average hourly value across all days [24]. IS values range from 0 to 1 with higher values representing greater regularity.

### 4.3. Missing Data Simulation and Imputation

For each participant, actigraphy timeseries data were masked with zeros or Not a Number values (NaNs) to simulate bouts of non-wear and missing data at multiple times of day, varying both bout duration and duration between multiple bouts (detailed in 4.3.1). Three methods of imputation were used to statistically replace missing data: linear interpolation, mean ToD imputation, and median ToD imputation (detailed in 4.3.2). IV and IS were calculated for each individual with each mask tested. Bland–Altman plots were generated using the masked and imputed IV and IS values as compared to the full data values. Mean, standard deviation, and slopes extracted from the Bland–Altman plots were used to generate heatmaps spanning the masking conditions (Figure 1B) (detailed in 4.4.). All data processing was done using MATLAB (R2020b, Mathworks, Natick, MA, USA).

#### 4.3.1. Masking

A series of masks were generated to simulate missing data by artificially replacing data with zeros or NaNs to represent possible phenotypes of missing data. Replacement with zeros is representative of the sensor being removed from the wrist but still collecting data. Replacement with NaNs is representative of the sensor turning off and collecting no data. Single bouts of missing data were simulated by varying the bout duration from 1 to 23 h in 2 h increments and the bout starting time across the day in 2 h increments (Figure 6A,B). We also examined instances in which we simulated multiple bouts of missing data on a single day and bouts missing at similar times on consecutive days. The selection of the multiple bout scheme was informed by missing data trends in the UK Biobank dataset. Missing data patterns within the complete UK Biobank dataset (Figure S1) indicated that two bouts of missing data were most common. The harmonic means of the duration of the first and second missing data bouts indicated representative durations of 113 min and 136 min, respectively (Figure S1). The start time of the first bout and the duration between the two bouts were varied. Start time was varied in two-hour increments across the day and week and the duration between gaps was varied in two-hour increments from 3 to 47 h (Figure 6C,D).

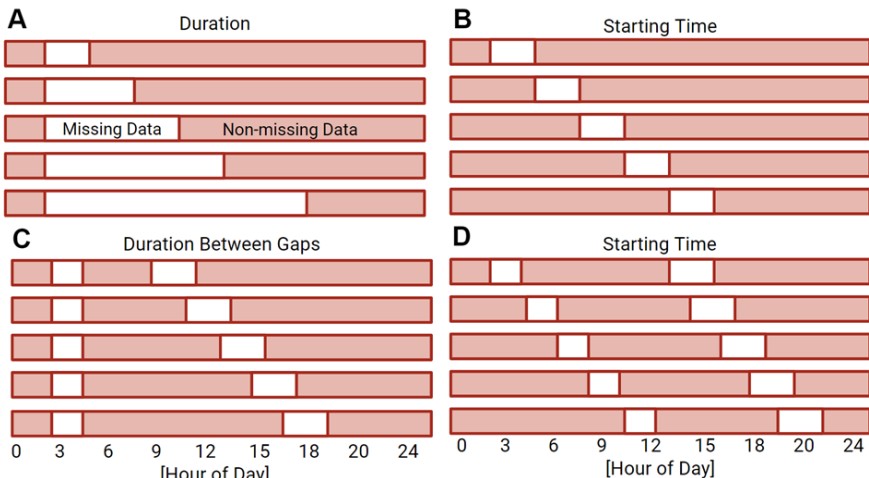

**Figure 6.** Mask overview. Data were systematically removed in single gaps at various durations (**A**), as well as single gaps starting at various times (**B**), while multiple gaps of missing data were varied in duration between gaps (**C**), as well as gap start time (**D**).

#### 4.3.2. Imputation Methods

Three common methods were selected to impute data, including linear interpolation, mean ToD imputation, and median ToD imputation. In linear interpolation, missing data are replaced with a line with slope and intercept set by the surrounding non-missing points and was calculated as:

$$a_{linear}(t) = \frac{a_{end+1} - a_{start-1}}{t_{end+1} - t_{start-1}} * (t - t_{start-1}) + a_{start-1} \tag{3}$$

where $a$ is the actigraphy value, $a_{linear}$ is the imputed actigraphy value, $t$ is the time, and *start* and *end* correspond to the start and end of the missing data segment, respectively. Note that imputation was performed only in the range of the missing data segments.

Mean ToD imputation, which is commonly used and is incorporated into the Biobank Accelerometry Analysis Python package, relies on the mean of the data on non-missing days at the corresponding timepoints of missing data to impute and is calculated as:

$$a_{mean}(t) = \frac{1}{N-1} \sum_{i=1}^{N} a_{i, \text{ToD}(t)} \tag{4}$$

where $a_{mean}$ is the mean ToD imputed actigraphy value, $N$ is the number of instances of time of day, ToD, corresponding with time, $t$. Imputation was performed in the range of the missing data gap and without including the missing data value in the mean calculation.

Median ToD imputation was calculated similarly to mean ToD imputation but uses the median of the non-missing days at the corresponding timepoints and is less sensitive to outliers. Median ToD imputation was calculated as:

$$a_{median}(t) = a_{\text{ToD}(t)} \left[ \frac{N-1}{2} \right] \tag{5}$$

where $a_{median}$ is median ToD imputed actigraphy value. Imputation was performed in the range of the missing data gap and without including the missing data value.

Generally, the imputation methods are statistical timeseries gap filling methods and as such are not identical to the data that have been masked (Figure 7). Linear interpolation is most sensitive to the values surrounding the missing data but does not consider data from other days without missing data (Figure 7B). Mean ToD imputation (Figure 7C) has greater sensitivity to outliers but less variability than median ToD imputation (Figure 7D). It is important to note that these imputation methods are intended to improve estimates of IV and IS rather than replace missing timeseries data.

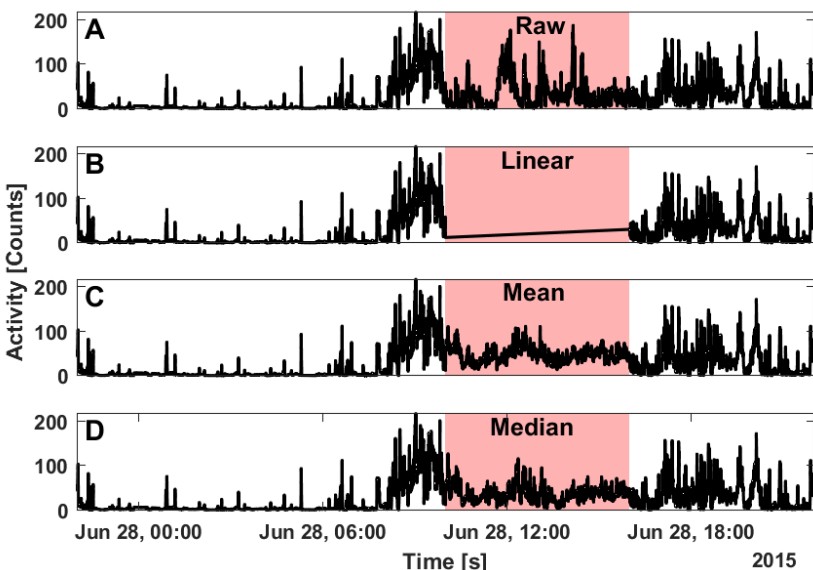

**Figure 7.** Example of a segment with complete data (**A**), and linear interpolation data (**B**), mean ToD imputed data (**C**), and median ToD imputed data (**D**) on 5 h of missing data starting at 10 am. Linear interpolation (**B**) is highly dependent on the values surrounding the gap, mean ToD imputation (**C**) has more smoothing than median ToD imputation (**D**); each of the imputation methods are statistical and do not perfectly represent the true data (**A**).

*4.4. Bland–Altman Plots and Heat Maps*

Bland–Altman plots were used to assess the impact of missing data and the performance of the imputation methods on the calculation of IS and IV. The mean, standard deviation, and slope from each of the Bland–Altman plots are presented as heat maps for clarity on the impact of missing data timing and duration on estimation of IV and IS. Bland–Altman plots were generated using the negative controls (masked or imputed) IV and IS values compared to the positive control (unadulterated data). The difference from the positive control was plotted against the average between the two compared measures for each condition (Figure 8). The mean difference, 1.96 × standard deviation, and the slope were extracted from the Bland–Altman plots for each of the masking conditions and imputation method. Heat maps were generated across all days of the week for both single and multiple masking conditions.

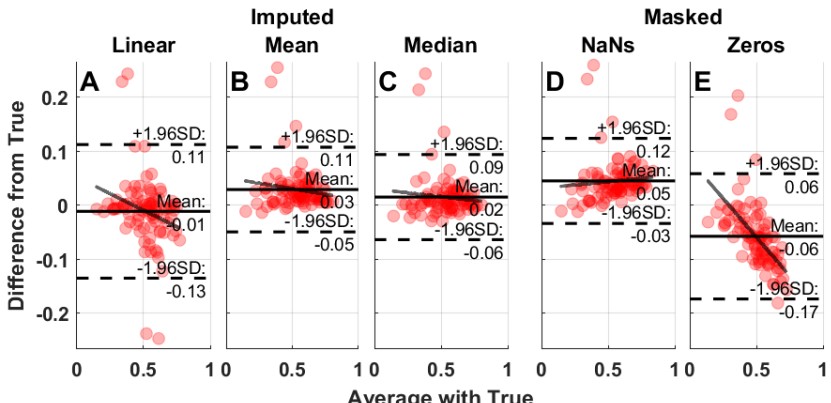

**Figure 8.** Sample Bland–Altman plots for IS masked with a single 5 h gap starting at 10 am and imputed. The solid black line depicts the mean, while dotted lines indicate ±1.96 × standard deviation and the gray line represents the linear fitted slope. Performance of linear interpolation (**A**), mean imputation (**B**), median imputation (**C**), data masked with NaNs (**D**), and data masked with zeros (**E**) are presented.

**Supplementary Materials:** The following supporting information can be downloaded at: https://www.mdpi.com/article/10.3390/clockssleep4040039/s1. Supplementary Materials contain histograms of non-wear episodes in the general population and heatmaps of Bland–Altman plot means, 1.96*standard deviation, and slopes for all days of the week. Code supporting this project can be found at https://github.com/ZeitzerLab/IVIS_Imputation (made available on 26 September 2022).

**Author Contributions:** Conceptualization, L.W., R.L. and J.Z.; methodology, L.W., D.C., R.L. and J.Z.; software, L.W. and D.C.; validation, L.W., R.L. and J.Z.; formal analysis, L.W.; investigation, L.W.; resources, J.Z.; data curation, L.W.; writing—original draft preparation, L.W., R.L. and J.Z.; writing—review and editing, L.W., R.L. and J.Z.; visualization, L.W. and R.L.; supervision, J.Z.; project administration, J.Z.; funding acquisition, N/A. All authors have read and agreed to the published version of the manuscript.

**Funding:** This research received no external funding.

**Institutional Review Board Statement:** Not applicable.

**Informed Consent Statement:** Not applicable.

**Data Availability Statement:** This research has been conducted using data from UK Biobank, a major biomedical database. Data that contributed to this research can be requested at https://www.ukbiobank.ac.uk/ (accessed on 16 September 2020).

**Acknowledgments:** Some of the computing for this project was performed on the Sherlock cluster. We would like to thank Stanford University and the Stanford Research Computing Center for providing computational resources and support that contributed to these research results. This research has been conducted using the UK Biobank Resource under Application Number 63099.

**Conflicts of Interest:** The authors declare no conflict of interest.

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
