# Peer review of "The Impact of Missing Data and Imputation Methods on the Analysis of 24-Hour Activity Patterns"

_2624-5175, doi:10.3390/clockssleep4040039_

Round 1
Reviewer 1 Report
The manuscript by Weed et al., seeks to assess the best method for compensating for missing data in actigraphy data. The evaluation takes the form of comparing masked (NaNs, and zeros) and imputed data. Overall the assessment of the results is well explained, and the final data is communicated effectively. I broadly agree with the manuscripts conclusions.
However, I do not agree with the way some of the manuscript is structured and presented.
Firstly, the choice of interpolation method needs to be better explained. It is obvious that a linear interpolation is not going to be suitable for this problem, and the poor performance of this imputation method in figures 1 and 2 is hardly surprising. A justification for including linear interpolation, e.g. as a negative control, should be included. For example Morelli et al 2019, used quadratic and cubic spine interpolation for a similar problem, which didn't perform better. Whatever the justification, it should be stated clearly, as the median and mean values are being compared to this interpolation.
Secondly, some further explanation for why both NaNs and Zeros were used for masking is required, and the logic of lumping these two together made evident. My assumption, based on similar data I have seen, is that a wearable device logging data when removed will record a zero. Therefore masking with zeros is the default type of "missing" data, although the term missing data should perhaps be re-evaluated here. Really, we have three different types of data presented. "zero record" data, data with missing values "NaN", and imputed values. This separation is important, as in many cases the masking of zero record data with missing values seems to perform as well as the extra work of imputing, which is an easy fix to data in most programming languages. It needs to be made clear why, considering how well NaN's perform, why interpolation is even needed.
There are some areas in the introduction where more detail is needed. For example, the paragraph in introduction:
"Several methods of timeseries imputation have been used to fill missing actigraphy 54
data ranging from methods relying exclusively on data surrounding the gap such as sim- 55
ple linear interpolation, methods relying on data from other days during the same time of 56
day such as time-of-day-based mean and median imputation, and more sophisticated ap- 57
proaches leveraging larger datasets such as deep learning methods [20]."
Needs further citation. I would like to see individual citations for each method, hopefully recent papers performing similar actigraphy data.
No scripts are provided for the analysis, so understanding the results presented is challenging. However, the individual plots in the SI material was appreciated, and were useful to see the underlying data.
Overall, I would be supportive for publication, with the following caveats and significant revisions to the manuscript. Specifically;
1) In the introduction more detail needs to be given to explain the problem, and specifically the zero values problem.
2) Further rationale needs to be given for the types of interpolation chosen, and if unable to justify then the authors should consider other methods.
3) Provide greater citations in the introduction surrounding time series imputation
4) A fuller comparison of NaN vs Zeros is included in the results (or discussion), as relevant.
4) I would have liked to have been given access to the underlying scripts, as this would have helped explain how the heatmaps were built. I would but reluctant to endorse this for publication without the Matlab files being provided. Given the utility of this analysis it is important to allow others to attempt the same imputation.
Reviewer 2 Report
Summary
The authors of the manuscript aim to determine the impact of missing data on the calculation of interdaily stability (IS) and intradaily variability (IV) using actigraphy timeseries datasets. They masked segments of week-long actigraphy data with zeros and “not a number” (NaNs) at different timings and for different durations. IS and IV were calculated using the true and masked data sets and compared. Effectiveness of imputation methods (linear interpolation, mean time of day and median time of day) was also evaluated. The results showed an impact of missing data for longer durations and specific timings, as well as more accurate calculation of IS and IV from incomplete data sets using the median time of day imputation method. These findings are novel and will have important implication for the calculation of IS and IV using incomplete data sets.
General concept comments
There are a few run on sentences that while clear, make the article ore complicated than necessary to read. (Ex. Lines 113-117)
Specific comments
· I believe line 80 should read “has an increased magnitude”.
· The figures appear blurry. The image quality should be improved.
· Lines 111-112: “When masked with NaNs compared to the complete dataset, IV showed little deviation in mean.” Perhaps this is my own lack of understanding, but I’m not sure this sentence is clear. IV showed little deviation in mean what?
· Line 113: I believe the word “correct” should be “corrected”.
· In the subject characteristics section of the results, it says the participants’ age ranged from 47-77 years old, however, later in the methods, it says the participants in the Biobank used were between 40-69 years of age. It seems there is a discrepancy here.
· In Figure 5, the flow chart says 19,747 participants were excluded, but the figure description says 19,748.
· What were the ranges of IV and IS in your study population? Would having a lower IV or greater IS value effect the impact of missing data?
Reviewer 3 Report
This study aims to compare the imputation methods for data lacking in the activity recording data. The conclusion, that median ToD imputation is the best, is clear and provides useful suggestions for future data analysis.
Major comment.
The mechanism of how median imputation worked better than mean imputation is not so much discussed. I guess that the mean imputation is largely affected by outlier data on the other day. Is it true? Are there other reasons?
Minor comment
The resolution of the Figure5 is too low to read.
Round 2
Reviewer 1 Report
The authors have addressed all of my concerns clearly, and have made all the revisions required. The code is supplied in a linked GitHub repository. I support this work for publication in its current form.